# Socio-Demographic, Clinical, and Mortality Differences between HIV-Infected and HIV/HTLV-1 Co-Infected Patients in Peru

**DOI:** 10.3390/pathogens12070869

**Published:** 2023-06-24

**Authors:** Maria Pia Amanzo-Vargas, Tessy Arellano-Veintemilla, Elsa González-Lagos, Juan Echevarría, Fernando Mejía, Ana Graña, Eduardo Gotuzzo

**Affiliations:** 1Facultad de Medicina, Universidad Peruana Cayetano Heredia, Lima 15102, Peru; maria.amanzo.v@upch.pe (M.P.A.-V.); tessy.arellano.v@upch.pe (T.A.-V.); elsa.gonzalez@upch.pe (E.G.-L.); juan.echevarria.z@upch.pe (J.E.); fernando.mejia.c@upch.pe (F.M.); 2Instituto de Medicina Tropical “Alexander von Humboldt”, Universidad Peruana Cayetano Heredia, Lima 15102, Peru; ana.grana@upch.pe; 3HIV Program, Hospital Cayetano Heredia, Lima 15102, Peru; 4Facultad de Enfermería, Universidad Peruana Cayetano Heredia, Lima 15102, Peru

**Keywords:** HIV, HTLV-1, co-infection, mortality, survival, antiretroviral therapy, Peru

## Abstract

Background and aims: In Peru, the estimated prevalence of human immunodeficiency virus (HIV) and human T-lymphotropic virus-1 (HTLV-1) co-infection has been reported to be as high as 18%. Despite the endemicity of HTLV-1 in Peru, few studies have assessed the impact of HIV/HTLV-1 co-infection. Our study compared socio-demographic and clinical characteristics, and mortality rates between HIV-infected and HIV/HTLV-1 co-infected patients. Methods: We reviewed the medical records of patients aged 18 years and older belonging to the HIV and HTLV-1 cohorts in Lima during a 30-year period: 1989–2019. Each HIV/HTLV-1 co-infected patient was randomly matched with two HIV-infected patients with similar characteristics (same sex, age ± 5 years, and same year of HIV diagnosis). Allegedly co-infected patients without a confirmatory diagnosis of HIV and HTLV-1 were excluded. Most of the patients in the HIV-infected group did not have a negative test result for HTLV-1 infection, so we used two probabilistic sensitivity analysis models to correct for potential HTLV-1 exposure misclassification bias in the group of HIV-infected patients. Results: Of 162 patients enrolled, 54 were HIV/HTLV-1 co-infected and 108 were HIV-infected. The median age was 42 years (IQR = 34–51 years) and the majority were male (61.1%), single (44.4%), heterosexual (71%), born in Lima (58%), educated at the secondary school level (55.6%), and receiving antiretroviral treatment (91.4%). HIV/HTLV-1 co-infection was associated with an increased risk of death (HR: 11.8; 95% CI: 1.55–89.00; *p* = 0.017) while antiretroviral treatment was associated with a decreased risk of death (HR: 0.03; 95% CI: 0.003–0.25; *p* = 0.001). The overall mortality rate was 13.6 per 100 persons and the survival time for co-infected patients (median = 14.19 years) was significantly shorter than that of HIV-infected patients (median = 23.83 years) (*p* < 0.001). Conclusions: HIV/HTLV-1 co-infected patients had a significantly shorter survival time compared to HIV-infected patients, suggesting that the immune alterations caused by HTLV-1 in CD4 cell count may have contributed to late initiation of antiretroviral treatment and prophylaxis against opportunistic infections over the decades, and thus reducing their benefits in these patients.

## 1. Introduction

Human T-cell lymphotropic virus type 1 (HTLV-1) was the first human retrovirus to be described in the early 1980s and it is estimated to affect at least 5–10 million people worldwide [1,2]. Southwestern Japan, Australia, and several countries from Africa and the region of the Americas present some of the highest prevalence rates of HTLV-1 globally [3]. In South America and the Caribbean basin, the areas with the highest and most widespread prevalence of HTLV-1 are found in Brazil, Colombia, French Guiana, Guyana, Haiti, Jamaica, and Peru [3]. The estimated prevalence in the general population of Peru is 2.9% [4]. 

HTLV-1 infection is characterised by the proliferation and persistence of infected CD4^+^ T cell clones that predispose affected individuals to the development of inflammatory and malignant diseases, as well as infectious complications. Among the associated inflammatory diseases is HTLV-1-associated myelopathy or tropical spastic paraparesis (HAM/TSP), which is characterised by a progressive decrease in strength in the lower limbs, urinary incontinence, and urinary tract infections, with a severely deteriorated quality of life. Other inflammatory conditions associated with HTLV-1 infection are uveitis, arthropathy, Sjögren’s syndrome, thyroiditis, and polymyositis [5]. The principal associated neoplasm is adult T-cell leukaemia/lymphoma (ATL), which has a survival time of less than one year in its most aggressive presentations [6]. Moreover, immune disturbances from HTLV-1 infection predispose individuals to infectious complications such as strongyloides hyperinfection syndrome, scabies, infectious dermatitis, and tuberculosis [5].

The human immunodeficiency virus (HIV) also belongs to the family Retroviridae; although it shares the same in vivo tropism towards CD4^+^ T cells as HTLV-1, its main mechanism of spreading is through viral replication, cell destruction, and continuous infection of new lymphocytes [7]. The cytopathic effect of HIV markedly weakens cellular immunity and predisposes individuals to infections from other pathogens including *Mycobacterium tuberculosis*, *Mycobacterium avium complex*, *Cryptococcus neoformans*, *Pneumocystis jirovecii*, *Toxoplasma gondii,* and *Cytomegalovirus*, as well as neoplasms such as central nervous system lymphoma and Kaposi’s sarcoma [7].

HIV/HTLV-1 co-infection has been associated not only with a higher risk of developing neurological complications such as HAM/TSP and peripheral neuropathy [8,9,10,11], but also with a higher frequency of crusted scabies, strongyloidiasis, extrapulmonary tuberculosis, pneumonia, and oesophageal candidiasis compared to those infected with HIV [8,12,13,14]. Furthermore, several studies have described an accelerated clinical progression to advanced HIV disease—the acquired immunodeficiency syndrome (AIDS)—and shorter survival time in patients with HIV/HTLV-1 co-infection compared to HIV-infected patients without HTLV-1 [15]. It has been suggested that death was mainly due to AIDS-related conditions, a CD4 cell count of less than 100 cell/µL, and lack of antiretroviral therapy (ART) [16,17]. In contrast, other studies describe co-infection as not having a significant impact on clinical progression or risk of death [16,18]. These differences may be due to heterogeneity in the use of ART across studies, especially since eligibility used to depend on patients reaching a low CD4 cell count threshold.

Although HIV/HTLV-1 co-infection is frequently reported in South American countries such as Peru, few studies have determined the impact of their interaction on clinical progression. We sought to compare socio-demographic and clinical characteristics as well as mortality rates between patients with HIV infection and HIV/HTLV-1 co-infection who attended our clinics from 1989 to 2019 in Lima, Peru.

## 2. Materials and Methods

### 2.1. Study Design and Participants

We deployed a matched retrospective cohort study design with data that had previously been collected prospectively from two clinical cohorts. The primary exposure was HIV/HTLV-1 dual infection compared to HIV infection; the principal outcome was mortality. We reviewed medical records corresponding to patients aged 18 years and older from the HIV and HTLV-1 cohorts of the Instituto de Medicina Tropical “Alexander von Humboldt” (IMTAvH), which specialises in infectious and tropical diseases. The HIV cohort included HIV-infected patients enrolled in the National Health Strategy for the Prevention and Control of STIs, HIV and AIDS at Hospital Cayetano Heredia. In the HTLV-1 cohort, patients with HTLV-1 confirmatory tests who attended the HTLV-1 unit at the same hospital were included and those with a positive result for HTLV-2 in the Western blot test were excluded. The acceptance rates for participation in the HIV and HTLV-1 cohorts were close to 80% and 85%, respectively. Once informed consent for entry into the respective IMTAvH cohort was obtained, a complete medical history, laboratory tests, and periodic follow-up were carried out and documented in the medical records of each cohort. We identified 75 medical records from HIV/HTLV-1 co-infected patients who were included in the HTLV-1 cohort from 1989 to 2019. Of this group, 12 patients were not eligible for inclusion (e.g., no confirmatory test done for HTLV-1 or HIV), leaving 54 eligible HIV/HTLV-1 co-infected patients. We then selected two HIV-infected patients for each of the 54 co-infected patients, such that 108 HIV-infected patients were included. These patients were selected randomly from a pool of HIV-infected patients that were matched with co-infected patients of the same sex, age at entry into the respective cohort (±5 years), and year of HIV diagnosis. 

### 2.2. Procedures

We examined the medical records of 162 patients in the two clinical cohorts to obtain information regarding socio-demographic, clinical, and laboratory characteristics, as well as vital status. Confirmatory diagnoses of HIV and HTLV-1 were established by one of the following positive tests: Western blot, indirect immunofluorescence assay, or polymerase chain reaction; or two positive enzyme immunoassays evidenced in the medical records. The first viral load (VL) and CD4 cell count obtained within six months of HIV diagnosis were the initial values. An undetectable VL corresponded to a value that was below the detection limit set by the laboratory for each year. In patients classified as HIV-infected, most of them did not have a negative test result for HTLV-1 infection, leading us to conduct sensitivity analyses as described below. We assessed vital status at the end of the follow-up period (30 May 2022) and, where applicable, the date of death through the National Registry of Identification and Civil Status (RENIEC) documented in the medical records. 

### 2.3. Statistical Analysis

For categorical variables, frequencies and percentages were estimated, while for numerical variables with non-normal distributions, the median and interquartile range (RQ: p25–p75) were calculated. To determine the association between independent variables and mortality, Pearson’s Chi-square and Fisher’s exact tests were used for categorical variables, and Mann–Whitney U test was used for numerical variables. Bivariate analysis was omitted for variables with more than 10% of values missing, including HIV viral load, CD4 count at baseline and at the last visit, and prophylaxis against opportunistic diseases. For univariate and multivariate analysis, a Cox regression model considering dynamic variables over time (date of HTLV-1 diagnosis, use and start date of ART, and HIV viral load value) was used to estimate crude and adjusted hazard ratios (HR). A value of *p* < 0.05 was considered statistically significant. The Kaplan–Meier curve was used to construct survival curves between the two groups.

Acknowledging a potential limitation in the classification of patients in the HIV-infected group, a sensitivity analysis was run under the probabilistic model, which was especially recommended to account for potential exposure misclassification bias [19]. Since a sensitivity analysis could not be run for the original time-varying survival model, it was adjusted for the basic cohort study model and relative risk calculation. Two models were thus run, each with 40,000 simulations, and it was considered that the error in sensitivity and specificity of classification of the HIV-infected group could range from 10% (high error rate) to 90% (low error rate), or a more conservative range of 25–75%. The model did not consider the range 0–100% because there were eight patients with confirmed negative HTLV-1 tests in the HIV-infected group. Statistical analyses were performed using STATA v17.0 (StataCorp; http://www.stata.com, accessed on 15 October 2022).

### 2.4. Ethical Review

In this study there was no contact with patients as it was based exclusively on retrospective review of medical records. The study was reviewed and approved by the Institutional Ethics Committee of Universidad Peruana Cayetano Heredia (approval number: 200-01-21).

## 3. Results

Of 162 patients selected and reviewed for this study, 54 were HIV/HTLV-1 co-infected and 108 were HIV-infected (Figure 1). The median age was 42 years (IQR 34–51 years); most patients were male (61.1%), single (44.4%), heterosexual (71%), born in Lima (58%), educated at the secondary school level (55.6%), and received ART (91.4%).

HIV/HTLV-1 co-infected patients were more likely to have been born outside of Lima (55.6%) compared to those infected with HIV (35%). Similarly, residing in the Andean region was more common in the group of co-infected patients (33.3%) compared to those infected with HIV (24.1%). Co-infected patients were more likely to have associated infections vs. HIV-infected patients: tuberculosis infection (40.7% vs. 26.0%, *p* = 0.05), herpes zoster disease (39.0% vs. 14.0%, *p* < 0.001), and hepatitis B (13.0% vs. 4.6%, *p* = 0.1). Table 1 summarises the socio-demographic, clinical, and laboratory characteristics of HIV/HTLV-1 co-infected and HIV-infected patients.

In the Cox regression model adjusted for sex, education level, marital status, tuberculosis infection, ART use, HIV stage, and baseline HIV viral load, co-infection with HTLV-1 was associated with an increased mortality (HR: 11.75; 95% CI: 1.55–89.00; *p* = 0.017) while ART use was associated with a decreased mortality (HR: 0.03; 95% CI: 0.003–0.25; *p* = 0.001). Table 2 presents factors associated with mortality in HIV/HTLV-1 co-infected and HIV-infected patients.

The overall mortality rate was 13.6 per 100 persons (22 reported deaths; 15 of 54 [27.8%] co-infected patients, 7 of 108 [6.5%] HIV-infected patients) and the survival time for co-infected patients (median = 14.2 years) was significantly shorter than that observed for HIV-infected patients (median = 23.8 years) (*p* < 0.001). Figure 2 shows the Kaplan–Meier survival curves for both groups.

Table 3 displays the two probabilistic sensitivity analysis models for the correction of HTLV-1 exposure misclassification bias in the group of HIV-infected patients. The first model has an error in sensitivity and specificity ranging from 10 to 90% while the second model ranges from 25 to 75%.

## 4. Discussion

In our study, HIV/HTLV-1 co-infected patients had a significantly shorter survival time compared to HIV-infected patients. This finding is consistent with previous studies that reported higher mortality rates among co-infected patients [13,20]. In a case–control study conducted at a referral centre in Salvador, Brazil, the survival time in co-infected patients (16.7 ± 0.7 years) was found to be significantly shorter compared to HIV-infected patients (18.1 ± 0.4 years) (*p* = 0.001) [17]. However, other studies did not find co-infection to be associated with an increased risk of death [16], shorter survival, or clinical progression to AIDS [18]. 

Although both viruses share similar transmission routes and risk factors [21,22], and the prevalence of their co-infection has been reported to be as high as 18.6% in specific Peruvian populations and regions [4,23], only a few studies similar to ours have been conducted. In our study, most of the co-infected patients were born in regions other than Lima, with one third of them being born in the Andean region, followed by the Amazon region and finally the rest of the coastal regions (*p* < 0.001). This finding is consistent with a study carried out in Lima, Peru, where 48% of patients infected with HTLV-1 were born in the Andean region and 68% had lived for prolonged periods in this region [24]. These findings suggest hyperendemicity of HTLV-1 infection in the Andean region. Data collected by the National Institute of Statistics and Informatics (INEI) from Peruvian nationwide surveys from 2007 to 2016 show a consistent pattern in which populations in the Andean and Amazonian regions have lower incomes and higher poverty than those living on the coast [25,26]. Moreover, the Andean and Amazonian regions are mainly rural, and their inhabitants tend to have less access to healthcare not only because of their distant and complex geographical location, but also because of pronounced socio-cultural barriers [27]. All of these factors may have influenced both access to healthcare and the quality of care they received.

Associated infections including herpes zoster disease and tuberculosis were more commonly seen in co-infected patients. A Peruvian study reported that, in women with herpes zoster disease, HTLV-1 infection is more frequent in those older than 50 years, especially when there is involvement of multiple dermatomes, whilst HIV infection was found mostly in women younger than 35 years [28]. With our sample size, the difference in tuberculosis was not statistically significant; however, the trend was consistent with a study in Guinea-Bissau that found HTLV-1 infection to be associated with an increased risk of developing tuberculosis in HIV-positive patients [29]. In our country, there is a scarce literature on this topic even though both HTLV-1 and tuberculosis infections are endemic. In an earlier Peruvian study with a population similar to ours, HTLV-1 infection was strongly and independently associated with mortality among inpatients with tuberculosis (OR: 9.43; 95% CI: 2.19–40.6) [30]. Additional Peruvian data also supports the conclusion that HTLV-1 infection can increase an individuals’ susceptibility to developing active tuberculosis [31]. Screening for herpes zoster disease and tuberculosis in HTLV-1 infected individuals, and particularly in HIV co-infected ones, is advisable. 

For three decades since the advent of zidovudine monotherapy in 1987, CD4 cell count was used to determine ART initiation, under the assumption that drug side effects and risk of drug resistance should be avoided until ART was needed the most [32]. In Peru, the National Programme of Highly Active Antiretroviral Therapy started in 2004 with its first criterion for ART initiation being a CD4 cell count of < 200 cells/mm^3^, although initially more than half of the patients started ART at a cell count of < 100 cells/mm^3^ [33]. Late initiation of ART has been shown to be associated with increased risk of disease progression and mortality [34,35], and over the years it has been recommended to start at a higher CD4 cell count [35]. Since 2016, the World Health Organization has recommended treatment for all people living with HIV, regardless of CD4 cell count [35]; however, it was not until 2018 that this proposal was included in Peru’s national medical guidelines [36]. In our study, the proportion of ART use was significantly higher in co-infected patients compared to HIV-infected. Nonetheless, given the broad time frame for patient selection in our study, there is significant variability in the ART initiation criteria used among patients, as well as in their availability in our clinics. 

HIV/HTLV-1 co-infected patients have higher CD4 cell count due to clonal proliferation as part of the pathophysiological mechanism of HTLV-1 that leads to overproduction of CD4 T-cells; however, these cells have reduced functionality and do not confer immunological benefits [7,12,14,37,38]. Brites et al. demonstrated that deceased co-infected patients had significantly higher baseline CD4 cell count than HIV-infected patients with the same outcome [17]. Schechter et al. emphasized their concerns about the use of CD4 cell count as a criterion for ART initiation and prophylaxis against opportunistic infections (OIs) [12]. In our study, we found that HIV/HTLV-1 co-infected patients had a higher baseline CD4 cell count and, in the multivariate Cox regression analysis, that this co-infection significantly increased mortality, whilst ART use decreased mortality. Using CD4 cell count to define the timing of ART initiation and prophylaxis represents a misinterpretation of the patient’s true immune status and risk of disease progression. Thus, a high CD4 cell count may have delayed the introduction of effective ART and prophylaxis against OIs, and probably increased mortality among HIV/HTLV-1 co-infected individuals; however, the extent to which it influenced our results was not assessed. 

A strength of our study is its sample size which was achieved in a highly HTLV-1 endemic region. One of the limitations of our study is that it was conducted in a single health centre in Lima; however, this is the principal national reference centre for the diagnosis and chronic care of patients with HTLV-1 infection. The retrospective nature of this study may have influenced the results as care and treatment for HIV-infected patients has changed over the years. It was not possible to confirm the absence of HTLV-1 infection in the group of patients classified as HIV-infected given the absence of free HTLV-1 screening in Peru and fiscal constraints experienced by the patients. To determine the impact of patients possibly infected with HTLV-1 in the group of HIV-infected patients, a probabilistic sensitivity analysis was performed and confirmed that our findings are not likely to be altered by a bias towards the null, namely the misclassification of a small number of co-infected patients in the HIV-only group. In addition, due to the lack of some clinical and laboratory data in the medical records, we suggest a more extensive clinical interview together with a wider availability of laboratory tests, as well as a thorough documentation of both, especially at the first contact with patients. Finally, despite the limitations of this study, our findings are relevant and unbiased, supporting those of other international research.

Our results demonstrate that ART not only reduces mortality but also that HIV/HTLV-1 co-infected patients have higher mortality rates and shorter survival times compared to HIV-infected patients. In our setting, the implementation of HTLV-1 screening tests should be prioritised at the national level, mainly in patients diagnosed with HIV, tuberculosis infection, herpes zoster disease, and pregnant women, especially in highly endemic regions of Peru. In patients with HIV/HTLV-1 co-infection, higher CD4 cell count could lead to misinterpretation of immune status and delay the initiation of prophylaxis against opportunistic infections, with a significant impact on morbidity and mortality. Finally, we recommend larger-scale prospective studies that cover more heterogeneous populations from different regions of Peru to broaden the understanding of the impact of HTLV-1 infection.

## Figures and Tables

**Figure 1 pathogens-12-00869-f001:**
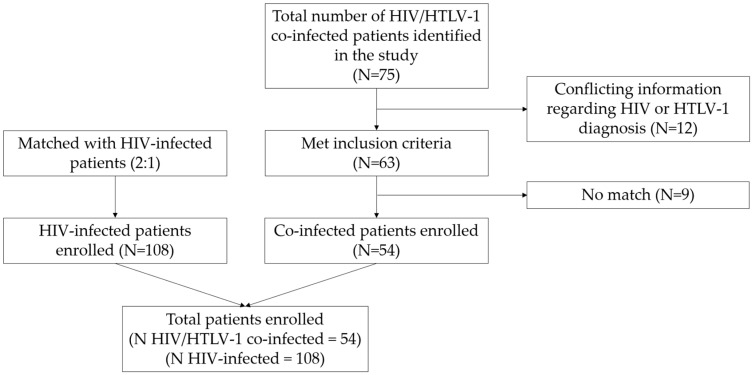
Flowchart for selection of HIV/HTLV-1 co-infected and HIV-infected patients within Peruvian clinical cohorts.

**Figure 2 pathogens-12-00869-f002:**
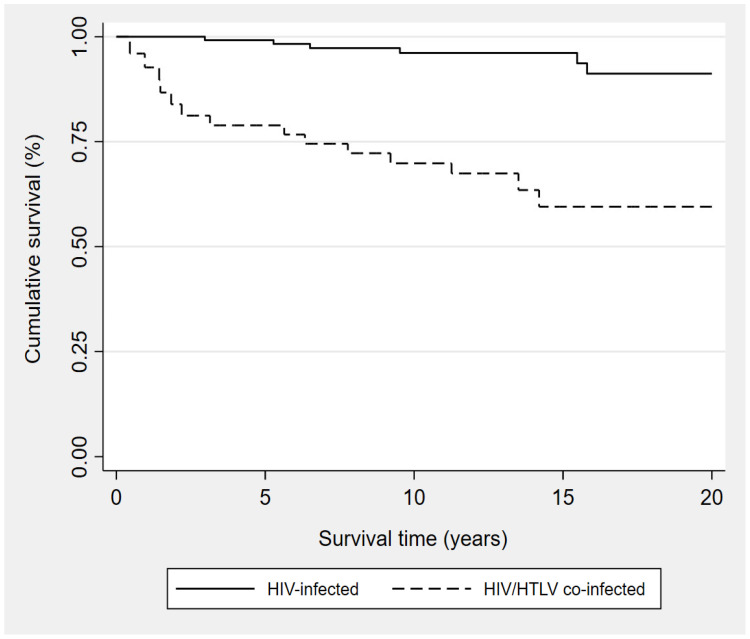
Kaplan–Meier survival curves from HIV diagnosis among HIV/HTLV-1* co-infected and HIV-infected patients. * Patients in the co-infected group were considered as such from the date of HTLV-1 diagnosis.

**Table 1 pathogens-12-00869-t001:** Socio-demographic, clinical, and laboratory characteristics of HIV/HTLV-1 co-infected and HIV-infected patients in Peru, matched on age, sex, and year of HIV diagnosis.

Characteristics	Total Sample	HIV-Infected	HIV/HTLV-1 Co-Infected	*p*
**Age at entry [median (IQR)]**	41.5 (34–51)	41 (35–51)	44 (33–51)	1.0 ^c^
**Sex, n (%)**				1.0 ^a^
Female	63 (38.9)	42 (38.9)	21 (38.9)	
Male	99 (61.1)	66 (61.1)	33 (61.1)	
**Marital status, n (%)**				<0.001 ^a^
Single	72 (44.4)	51 (47.2)	21 (38.9)	
Married/Cohabitant	49 (30.2)	40 (37.0)	9 (16.7)	
Separated/Divorced/Widowed	26 (16.0)	17 (15.7)	9 (16.7)	
Not specified	15 (9.3)	0 (0.0)	15 (27.8)	
**Sexual orientation, n (%)**				0.05 ^a^
Heterosexual	115 (71.0)	81 (75.0)	34 (63.0)	
Homosexual	26 (16.0)	13 (12.0)	13 (24.0)	
Bisexual	17 (10.5)	14 (13.0)	3 (5.5)	
Not specified	4 (2.5)	0 (0.0)	4 (7.4)	
**Birthplace, n (%)**				<0.001 ^b^
Coastal region: Lima	94 (58.0)	70 (65.0)	24 (44.4)	
Rest of coastal region	17 (10.5)	12 (11.1)	5 (9.3)	
Andean region	44 (27.1)	26 (24.1)	18 (33.3)	
Amazon region	7 (4.3)	0 (0.0)	7 (13.0)	
**Level of education, n (%)**				<0.001 ^a^
None/Primary education	19 (11.7)	11 (10.2)	8 (14.8)	
Secondary education	90 (55.6)	68 (63.0)	22 (40.7)	
Higher education	38 (23.5)	29 (26.8)	9 (16.7)	
Not specified	15 (9.3)	0 (0.0)	15 (27.8)	
**Employment status, n (%)**				0.8 ^a^
Employed	84 (52.0)	56 (51.8)	28 (51.8)	
Unemployed	76 (47.0)	52 (48.1)	24 (44.4)	
Not specified	2 (1.0)	0 (0.0)	2 (3.7)	
**Year of HIV diagnosis, n (%)**				1.0 ^b^
1998–2003	12 (7.4)	8 (7.4)	4 (7.4)	
2004–2010	93 (57.4)	62 (57.4)	31 (57.4)	
2011–2019	57 (35.2)	38 (35.2)	19 (35.2)	
**Hepatitis B virus infection, n (%)**				0.1 ^b^
Yes	12 (7.4)	5 (4.6)	7 (13.0)	
No	150 (92.6)	103 (95.4)	47 (87.0)	
**Sexually transmitted diseases** ***, n (%)**				0.30 ^a^
Yes	60 (37.0)	37 (34.3)	23 (42.6)	
No	102 (63.0)	71 (65.7)	31 (57.4)	
**Herpes zoster disease, n (%)**				<0.001 ^a^
Yes	36 (22.2)	15 (13.9)	21 (38.9)	
No	126 (77.8)	93 (86.1)	33 (61.1)	
**Tuberculosis** †**, n (%)**				0.05 ^a^
Yes	50 (30.9)	28 (26.0)	22 (40.7)	
No	112 (69.1)	80 (74.0)	32 (59.3)	
**Prophylaxis for OIs** ◊**, n (%)**				
Yes	79 (48.7)	60 (55.5)	19 (35.2)	
No	46 (28.4)	31 (28.7)	15 (27.8)	
Not specified	37 (22.8)	17 (15.7)	20 (37.0)	
**Initial HIV stage** ◊^**, n (%)**				
1	9 (5.6)	7 (6.4)	2 (3.8)	
2	48 (29.6)	35 (32.1)	13 (24.5)	
3	26 (16.1)	19 (17.4)	7 (13.2)	
4	59 (36.4)	45 (41.3)	14 (26.4)	
Not specified	20 (12.3)	3 (2.8)	17 (32.1)	
**Initial HIV viral load** ◊**, n (%)**				
Undetectable	7 (4.3)	7 (6.5)	0 (0.0)	
Detectable	112 (69.1)	84 (77.8)	28 (52.0)	
Not specified	43 (26.5)	17 (15.7)	26 (48.0)	
**HIV viral load at last visit** ◊**, n (%)**				
Undetectable	126 (77.8)	95 (88.0)	31 (57.4)	
Detectable	20 (12.3)	13 (12.0)	7 (13.0)	
Not specified	16 (10)	0 (0.0)	16 (29.6)	
**Initial CD4 cell count (cells/mm^3^)** ◊**, n (%)**				
<200	81 (50.0)	66 (61.1)	15 (27.7)	
200–499	42 (25.9)	29 (26.8)	13 (24.1)	
≥500	8 (5.0)	4 (3.7)	4 (7.4)	
Not specified	31 (19.1)	9 (8.3)	22 (40.7)	
**CD4 cell count at last visit (cells/mm^3^)** ◊**, n (%)**				
<200	20 (12.6)	11 (10.2)	9 (16.7)	
200–499	48 (29.6)	37 (34.3)	11 (20.4)	
≥500	78 (48.1)	60 (55.6)	18 (33.3)	
Not specified	16 (10)	0 (0.0)	16 (29.6)	
**Use of ART, n (%)**				<0.001 ^b^
Yes	148 (91.4)	108 (100)	40 (74.1)	
No	14 (8.6)	0 (0.0)	14 (25.9)	
**Death during follow-up, n (%)**				<0.001 ^a^
Yes	22 (13.6)	7 (6.5)	15 (27.8)	
No	140 (86.4)	101 (93.5)	39 (72.2)	

* *Chlamydia trachomatis, Neisseria gonorrhoeae*, *Treponema pallidum*, and *Haemophilus ducreyi* infections, herpes and genital warts, pediculosis pubis, and inguinal buboes. † Pulmonary and/or extrapulmonary tuberculosis infection. ◊ Bivariate analysis was not performed due to 10–26.5% missing data classified as not specified. ^ WHO clinical staging of HIV/AIDS for adults and adolescents with confirmed HIV infection. ^a^ Pearson’s Chi-square test. ^b^ Fisher’s exact test. ^c^ Mann–Whitney U test.

**Table 2 pathogens-12-00869-t002:** Factors associated with mortality in HIV/HTLV-1 co-infected and HIV-infected patients.

Characteristics	Crude HR	95% CI	*p*	Adjusted HR	95% CI	*p*
**Age at entry**	1.0	0.97–1.06	0.5			
**Sex**						
Female	Reference			Reference		
Male	1.2	0.47–3.21	0.7	0.2	0.02–1.96	0.2
**Marital status**						
Single	Reference			Reference		
Married/Cohabitant	1.2	0.40–3.78	0.7	0.5	0.05–4.31	0.5
Separated/Divorced/Widowed	0.6	0.12–3.02	0.5	0.2	0.01–3.60	0.3
Not specified	5.6	1.89–16.84	0.002	3.3	8.54 × 10^−20^–1.25 × 10^20^	1.0
**Sexual orientation**						
Heterosexual	Reference					
Homosexual	1.7	0.60–4.78	0.3			
Bisexual	0.3	0.04–2.62	0.3			
**Birthplace**						
Coastal region: Lima	Reference					
Rest of coastal region	0.6	0.12–2.94	0.5			
Andean region	0.5	0.16–1.57	0.2			
Amazon region	1.2	0.16–9.51	0.8			
**Level of education**						
None/Primary education	Reference			Reference		
Secondary education	0.9	0.18–4.14	0.9	0.9	0.06–14.44	1.0
Higher education	1.2	0.23–6.46	0.8	2.1	0.12–39.88	0.6
Not specified	5.4	1.08–27.16	0.04	0.8	1.76 × 10^−20^–3.38 × 10^19^	1.0
**Employment status**						
Unemployed	Reference					
Employed	0.9	0.37–2.28	0.9			
**Associated retroviral infection**						
HIV	Reference			Reference		
HIV/HTLV-1	6.8	2.7–17.0	<0.001	11.75	1.55–89.00	0.02
**Hepatitis B virus infection**						
No	Reference					
Yes	1.2	0.3–5.7	0.8			
**Sexually transmitted diseases** *						
No	Reference					
Yes	0.6	0.2–1.6	0.3			
**Herpes zoster disease**						
No	Reference					
Yes	1.4	0.5–3.6	0.5			
**Tuberculosis** †						
No	Reference			Reference		
Yes	1.1	0.5–2.8	0.8	0.30	0.03–3.03	0.3
**Prophylaxis for OIs**						
No	Reference					
Yes	0.6	0.2–1.9	0.4			
**Use of ART**						
No	Reference			Reference		
Yes	0.07	0.0–0.2	<0.001	0.03	0.003–0.25	0.001
**Initial HIV stage**						
1	Reference			Reference		
2	0.2	0.03–1.3	0.1	0.03	0.0009–0.95	0.05
3	0.4	0.06–2.5	0.3	0.03	0.0008–1.22	0.06
4	0.4	0.1–1.7	0.2	0.06	0.004–1.20	0.07
Not specified	2.0	0.5–8.6	0.3	0.61	0.015–24.53	0.8
**Initial HIV viral load** ^						
Undetectable	Reference			Reference		
Detectable	0.7	0.1–5.6	0.7	0.1	0.004–1.33	0.1
**HIV viral load at last visit**						
Undetectable	Reference					
Detectable	4.4	1.5–13.2	0.008			
**Initial CD4 cell count (cells/mm^3^)**						
<200	Reference					
200–500	0.7	0.2–2.5	0.6			
>500	5.1 × 10^−16^	0–	1.00			
**CD4 cell count at last visit (cells/mm^3^)**						
<200	Reference					
200–500	0.2	0.07–0.8	0.02			
>500	0.1	0.02–0.3	<0.001			

* *Chlamydia trachomatis*, *Neisseria gonorrhoeae*, *Treponema pallidum*, and *Haemophilus ducreyi* infections, herpes and genital warts, pediculosis pubis, and inguinal buboes. † Pulmonary and/or extrapulmonary tuberculosis infection. ^ WHO clinical staging of HIV/AIDS for adults and adolescents with confirmed HIV infection.

**Table 3 pathogens-12-00869-t003:** Probabilistic sensitivity analysis for the correction of HTLV-1 exposure misclassification bias in the group of HIV-infected patients.

	RR	95% CI
**Model: 10–90%**		
Original study	4.11	1.77–9.58
Sensitivity analysis	5.11	0.45–17.87
**Model: 25–75%**		
Original study	4.11	1.77–9.58
Sensitivity analysis	13.01	1.66–19.64

## Data Availability

Data are available upon request.

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
