# Peer review of "Socio-Demographic, Clinical, and Mortality Differences between HIV-Infected and HIV/HTLV-1 Co-Infected Patients in Peru"

_pathogens, 2023, doi:10.3390/pathogens12070869_

Round 1

Reviewer 1 Report

The manuscript entitled "Socio-demographic, clinical, and mortality differences between HIV-infected and HIV/HTLV-1 co-infected patients in Peru", from Amanzo et al, used a retrospective cohort to evaluate the frequency and impact of HIV-HTLV coinfection on AIDS progression. They used epidemiological records to perform the evaluation, by chart review of  30 years.

They detected  a higher prevalence of men, higher risk of herpes zoster and hgiher frequency of ISTs among coinfected than HIV infected ones. They also found a higher mortality rate and shorter survival time for coinfected subjects. They also detected a protective effect of ART, as expected.

Some specific comments:

There is a problem with references: for instance, reference 33 is not included in the reference list, but cited in text. In addition, some references do not fit with the citations in text. Reference 8 is cited as a support of shorter survival, but the original study is a cross-sectional one focused on immune activation and its potential implications.At lines 222-224 the cited reasons for early treatment of HIV are not adequate, as early treatment is a piece of TASP strategy and protective against AIDS-related and not related morbi-mortality. In addition, the statement that ART increases mortality is not true. The sentence structure must be modified to make it clear.

The limitations of the study are not well emphasized, authors should include the retrospective design and the long period of time for patients inclusion. ART and HIV care significantly changed in the last 3 decades, what could impact the finding of such long study, but this is not cited by authors. Providing the time of entry in the cohort for patients would be of help and would make easier to understand how time sensitive issues (like ART!) could afect the results.

English style must be revised, some parts of text are hard to understand. I suggest to avoid terms like "dramatic" or "dramatically", they do not add to text clearness and means nothing, in a scientific point of view.

The text is confuse in some parts, and English should be revised for clearness sake.

Reviewer 2 Report

This is an intriguing study by Amanzo et al. that focused on HIV/HTLV-1 co-infected patients who had a significantly shorter survival time compared to HIV-infected patients, which suggests that HTLV-related immunological perturbations may diminish the advantages of antiretroviral treatment in patients who are co-infected with HIV and HTLV-1.  The size of our study's sample population in an area with a high HTLV-1 prevalence is one of its strengths. Although there were a few limitation,  probabilistic sensitivity analysis was helpful to confirm the results. 

Reviewer 3 Report

The manuscript submitted by Amanzo et al. compares sociodemographic and clinical characteristics and mortality rates between HIV-infected and HIV/HTLV-1-coinfected patients in Peru. The data on mortality in patients coinfected with HIV/HTLV are of interest; however, the manuscript needs to be revised before it is accepted for publication.

The aim of the study needs to be clearly stated. Was it just to compare sociodemographic and clinical variables and mortality rates? Or did the authors want to verify whether coinfection between HTLV and HIV increases mortality?

The authors should better justify in the introduction why mortality associated with HIV/HTLV coinfection should be studied. What was the authors' hypothesis?

In line 51, the authors state, "Immune disorders due to HTLV-1 infection predispose to infectious complications such as Strongyloides stercoralis hyperinfection, scabies, infectious dermatitis, and tuberculosis" Expalin why and how coinfection between HTLV and HIV would lead to higher mortality in coinfected individuals?

The authors do not cite a recent systematic review that compared the clinical and laboratory outcomes of persons coinfected with HTLV and HIV with those of HIV monoinfected persons. This systematic review concludes that patients coinfected with HIV/HTLV have shorter survival, higher mortality rates, and more rapid progression to death than patients infected with HIV alone.

"Montaño-Castellón I, Marconi CSC, Saffe C, and Brites C (2022) Clinical and Laboratory Outcomes in HIV-1 and HTLV-1/2 Coinfection: A Systematic Review. Front. Public Health 10:820727. doi: 10.3389/fpubh.2022.820727"

Regarding methodology - it would be important to better characterize the study site and cohorts of the Institute of Tropical Medicine 'Alexander von Humboldt' (IM -80 TAvH).

A larger proportion of co-infected patients were from the Andes and the Amazon. What does this mean in terms of higher mortality? Do these patients have poorer access to health care or antiretroviral treatment? Do they have lower incomes? The authors should discuss this finding.

The authors conclude in the abstract that "HIV/HTLV-1-coinfected patients had a significantly shorter survival time compared to HIV-infected patients, suggesting that HTLV-related immune perturbations may reduce antiretroviral benefits in co-infected patients."

Do the authors really think immune alterations are responsible for the higher mortality? Coinfected patients appear to receive HIV diagnosis at a later time and have higher rates of unknown viral load and CD4 levels. In addition, 25% of them do not receive antiretroviral treatment. How do the authors explain these differences between patients?

At the end of the discussion, they state, "Our results show that HIV/HTLV-1 co-infected patients had higher mortality and shorter survival time compared to HIV-infected patients and that ART reduced mortality." An important finding of this manuscript was that co-infected patients have a lower proportion of trated patients compared with HIV-infected patients. In addition, at the time of diagnosis, almost one-third of co-infected patients had unknown CD4, viral load, or stage data from AIDS. What do the authors recommend to change this scenario?

Round 2

Reviewer 1 Report

I think authors provided he required corrections in the MS. 

Although most of the text was improved in comparison to the original version. 

Reviewer 3 Report

No comments.